

# Automatic pulmonary artery-vein separation in CT images using a twin-pipe network and topology reconstruction

Lin Pan[1], Xiaochao Yan[1], Yaoyong Zheng[1], Liqin Huang[1], Zhen Zhang[1], Rongda Fu[1], Bin Zheng[2] and Shaohua Zheng[1]

[1] College of Physics and Information Engineering, Fuzhou University, Fuzhou, Fujian, China
[2] Key Laboratory of Cardio-Thoracic Surgery, Fujian Medical University, Fuzhou, Fujian, China

## ABSTRACT

**Background:** With the wide application of CT scanning, the separation of pulmonary arteries and veins (A/V) based on CT images plays an important role for assisting surgeons in preoperative planning of lung cancer surgery. However, distinguishing between arteries and veins in chest CT images remains challenging due to the complex structure and the presence of their similarities.

**Methods:** We proposed a novel method for automatically separating pulmonary arteries and veins based on vessel topology information and a twin-pipe deep learning network. First, vessel tree topology is constructed by combining scale-space particles and multi-stencils fast marching (MSFM) methods to ensure the continuity and authenticity of the topology. Second, a twin-pipe network is designed to learn the multiscale differences between arteries and veins and the characteristics of the small arteries that closely accompany bronchi. Finally, we designed a topology optimizer that considers interbranch and intrabranch topological relationships to optimize the results of arteries and veins classification.

**Results:** The proposed approach is validated on the public dataset CARVE14 and our private dataset. Compared with ground truth, the proposed method achieves an average accuracy of 90.1% on the CARVE14 dataset, and 96.2% on our local dataset.

**Conclusions:** The method can effectively separate pulmonary arteries and veins and has good generalization for chest CT images from different devices, as well as enhanced and noncontrast CT image sequences from the same device.

## INTRODUCTION

Lung cancer is a significant global health concern with high morbidity and mortality rates (*Siegel et al., 2022*). In 2020, an estimated 19.3 million new cancer cases and nearly 10 million cancer deaths are expected worldwide, and lung cancer remains the leading cause of cancer deaths, accounting for around 18% of all cancer deaths. Computed tomography (CT), with its high resolution and ability to show the fine structures and density differences of lung tissues, has become a major tool for thoracic surgeons to screen, diagnose and treat diseases. Currently, in the clinic, surgeons generally manually reconstruct lung anatomy by

Corresponding author
Shaohua Zheng,
sunphen@fzu.edu.cn

using commercial medical software such as IQQA (*Xu et al., 2019*) and Mimics, which can then be used to analyze lung disease and the anatomical relationships among pulmonary arteries, veins, airways (*Guo et al., 2022*), and nodules (*Zheng et al., 2021*). However, manual reconstruction of arteries and veins is time-consuming and visually demanding due to the large quantity of CT data, which puts a heavy burden on surgeons. Moreover, the accurate separation of lung veins and arteries plays a crucial role in diagnosing and evaluating lung diseases such as lung cancer. Therefore, automatic and rapid vessel segmentation and separation of arteries and veins through CT scans is crucial for clinical diagnosis and can provide better survival prediction for lung cancer patients (*Haq et al., 2022*).

At present, the automatic separation of pulmonary arteries and veins from CT scans has become a popular topic in research. However, this task exist several difficulties, which can be attributed to the following reasons. Firstly, arteries and veins are indistinguishable due to their similar intensity values in noncontrast CT images. Secondly, the vessel tree structure is extremely complex and dense, with arteries and veins close to each other and intertwined. And finally, artifacts, partial volume effects, and patient-specific vessel tree structural abnormalities cause difficulty in A/V separation.

Recently, deep learning-based methods have demonstrated powerful feature extraction and analysis capabilities, and have achieved remarkable results in medical image analysis (*Abedalla et al., 2021*; *Nguyen et al., 2021*). With the development of deep learning, lung vessel segmentation has been widely explored (*Nam et al., 2021*). Although these studies have had a certain theoretical basis and achieved satisfactory results, there are still some intractable problems in the results after the segmentation of vessels. For example, the pulmonary vessel is a typical tree-like structure, with thick and thin vessels connected and numerous bifurcations. Existing deep learning methods rarely consider the topological information of the vessel structure, which can result in the separated arteries and veins being discontinuous or with breaks in the vessel branches. Additionally, tubular vessels in the lungs are small and have thin walls, making them difficult to detect and classify accurately using deep learning methods. These vessels are often mistaken for noise or artifacts in the image, leading to misclassification or missed detection. Furthermore, due to the similarity in size, shape, and appearance of arteries and veins, they may be difficult to distinguish in medical images and can be confused with each other after the A/V separation.

To address these concerns, we propose a pulmonary A/V separation method using topology information and a twin-pipe network. Our contributions are summarized as follows:

1) To address the discontinuity problem in A/V trees after separation, a vessel tree topology is constructed by combining scale-space particles and multi-stencils fast marching (MSFM) to ensure the continuity and authenticity of the topology reconstruction.

2) To address the poor separation effect of tubule vessels, a twin-pipe network is proposed to learn the multiscale differences between arteries and veins, as well as the

characteristics of small arteries that are closely accompanied by bronchi, given that arteries and bronchi tend to accompany each other.

3) To address the intertwining of arteries and veins, a topology optimizer is designed that considers both interbranch and intrabranch topological relationships to optimize the A/V classification results.

Our method is evaluated on our private database and on the public dataset CARVE14 (*Charbonnier et al., 2015*). Furthermore, we evaluated our method on another device and a CT scan of an arterial enhancement model to demonstrate its generalizability across different devices.

The rest of this article is organized as follows. In "Methods" shows the proposed approach to solve the A/V separation problem in noncontrast CT images. "Experiments" mainly introduces the experiment methods, including data sources, experimental setup details, and evaluation metrics. Then, in "Results", the experimental results, the ablation experiment, and the generalization experiment are presented. "Discussion" elaborates on the discussion. Finally, "Conclusion" concludes this article.

## RELATED WORKS

Although pulmonary A/V separation is a difficult problem, many scholars have proposed methods in the past decade. Studies have shown that the vasculature within the lungs is highly variable, but some inherent anatomical properties are usually present. One such property is that arteries in the lungs typically follow the bronchial tree, while veins tend to run in the interstitial spaces between the branches. As shown in Fig. 1, the arteries with accompanying bronchi are not evident when the vessels are near the hilum of the lung. As the arteries move away from the hilum, the bronchi begin to follow the arteries closely. Some methods rely on bronchial features for A/V separation. For instance, *Tozaki et al. (2001)* used information about distances between vessel segments and bronchi to separate arteries and veins, and *Nakamura et al. (2005)* classified pulmonary arteries and veins based on the distance from the bronchi region to the vessel segment and the distance from the nearest interlobar to the vessel. Similarly, *Bülow et al. (2005)* based upon the fact that the pulmonary artery tree accompanies the bronchial tree, designed a method of "arterialness" to classify each vessel segment. These pulmonary A/V separation methods rely on the quality of airway segmentation, while the use of bronchial features to aid A/V separation is rarely applied in deep learning. Our method takes this into account.

Another type of A/V separation study focuses on the anatomical priors of the vessels themselves, using information on their connectivity to separate arteries and veins. For example, *Saha et al. (2010)* proposed a method to separate A/V of pulmonary using morphological features of vessels. The approach involves selecting seed points and utilizing fuzzy distance transformation for tracking to ensure vessel connectivity. Finally, arteries and veins points are separated through multiscale iterative growth. *Wala et al. (2011)* designed an automated trace-based separation scheme that tracked arterial seed points from the basal pulmonary artery region and detected bifurcations to separate the artery from other nearby isometric structures. *Park et al. (2013)* proposed a method to find

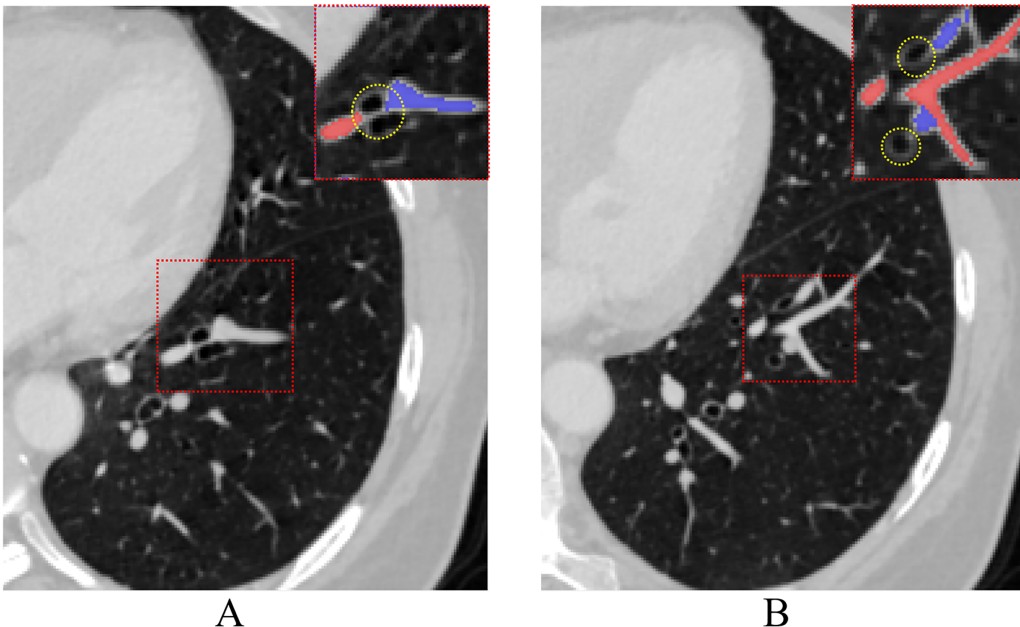

**Figure 1** **The process of pulmonary vessels moving away from the hilum accompanied by the bronchi on CT.** Vessels close to the hilum (A) and away from the hilum (B), red dashed rectangles highlight airways (yellow dotted circle) associated with arteries and veins at different levels.

voxel-based trees with minimal construction energy inside vessel segmentation and divided them into a group of subtrees that share the same A/V classification. Additionally, *Kitamura et al. (2016)* designed a method to classify pulmonary artery and vein voxels based on vascular connection information and energy minimization of high-order potentials. However, these methods are usually semi-automatic or dependent on specific CT quality. Besides, the use of the anatomical priors for the pulmonary A/V separation method is susceptible to limitations of the anatomical structures themselves, such as bronchial leakage, vessel adhesions and discontinuity.

Moreover, several new ideas for automatic A/V separation have been proposed. *Charbonnier et al. (2015)* proposed a method to convert vessel segmentation into a geometric graph representation. *Payer et al. (2016)* proposed an automated algorithm for A/V separation based on two integer programming. Through integer programming, parameter tuning was carried out to extract subtrees, and A/V separation was performed. The method used local information to detect the attached arteries and veins in the geometric graph to generate subgraphs. Then, the volume difference between arteries and veins was used for classification. *Park, Bajaj & Gladish (2018)* developed a method that utilizes sphere inflation tracking from the endpoint to the heart. This approach aims to differentiate between bifurcation and crossover points and automatically detect the pulmonary trunk. *Jimenez-Carretero et al. (2019)* proposed a new pulmonary A/V separation scheme, in which a specialized graph-cut method was designed to ensure the connectivity and consistency of the vessel subtrees. *Yu et al. (2021)* proposed a combined algorithm for pulmonary A/V separation based on a subtree relationship to separate the

adhesion points of the vessel tree structure. Generally, these methods are effective on the image with noise-free or uniform intensities but require complex parameter adjustment and parameter sensitive.

In recent years, many researchers have attempted to solve the problem of A/V separation using deep learning techniques, which offer better robustness and stability. *Nardelli et al. (2018)* proposed a convolutional neural network for A/V classification of pulmonary lobar vessel particles, an initial A/V classification is performed for each particle in the extracted vessel tree using CNN. The final classification result was obtained by a graph-cut strategy that combined connectivity and pre-classification information. *Zhai et al. (2019)* designed a network linking the CNN and graph convolutional network (GCN) to combine local image information with graph connection information to train the constructed graph, and the A/V separation results were obtained. Although U-Net is a common method in medical image segmentation, few works have been done in recent years for A/V separation. *Qin et al. (2021)* proposed learning tube-sensitive U-Net for lung A/V classification and anatomy prior of lung context map and distance transform map is combined to detect more arterioles and veins. *Heitz et al. (2021)* relies on three-paths 2.5D U-Net networks along axial, coronal, and sagittal slices to extract the tubular structures of the lung. These networks have partly solved the problems in pulmonary A/V separation, however, none of them consider the complex structures and topological connectivity of the lung vessels. Confusion and discontinuity of the pulmonary arteries and veins remained, especially in the hilar, adhesions, and terminal vessel regions.

## METHODS

The overall framework of the pulmonary A/V separation method in this article is shown in Fig. 2, including vessel tree topology extraction module, twin-pipe network, and topology optimizer. Portions of this text were previously published as part of a preprint (https://arxiv.org/abs/2103.11736).

In the vessel tree topology extraction module, the vessel tree is segmented by 3D U-Net from the chest CT images, and the topology is extracted from the vessel tree. Then, the distance transform is used to guide and compensate for the missing points. In the twin-pipe network, 3D patches are taken from each particle on the topological tree as the centers. The vessel segment and the tubule vessel are then trained separately to derive the preliminary classification results of the vessel particles. In the final optimizer module, the topology of the subtrees and their branches are extracted from the topological tree, the branch confidence is calculated, and subtrees are pruned to optimize A/V classification using topological connectivity. Finally, the arteries and veins of pulmonary topology are reconstructed, and the reconstructed A/V and separately segmented arteries and veins near the hilum are fused to obtain the final result.

### Vessel tree topology module

The proposed method begins with vessel tree segmentation. Vessel trees are extracted by using 3D U-Net with the patch of [256, 256, 16]. Then, a complete topological vessel tree is constructed by topological extraction of the vessel tree. Due to the applicability of the

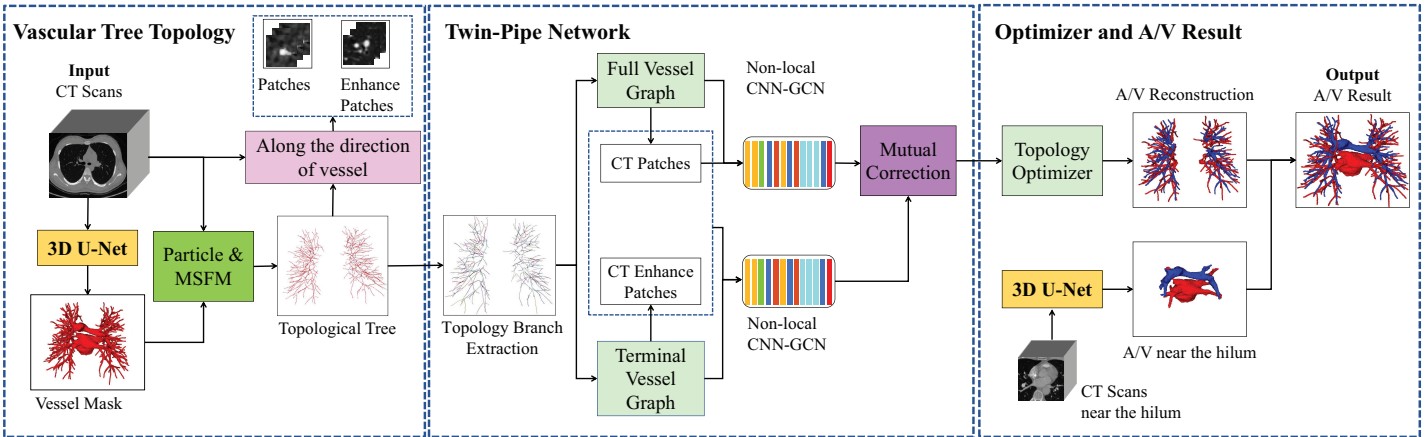

**Figure 2** Overview of the proposed method of pulmonary A/V separation including three modules: vessel tree topology extraction, twin-pipe network, and topology optimizer.

skeleton algorithm and the characteristics of the vessel tree itself, it is often not available to obtain accurate results of skeleton structure. To obtain a better topological veessel tree, we flexibly combine scale-space particles (*Estépar et al., 2012*) and the multi-stencils fast marching (MSFM) (*Liu et al., 2018*), superior to some existing special methods such as thinning, geometry, shortest path, *etc.*, the specific steps are as follows.

Firstly, based on the characteristics of the vessel tubular structure, scale-space particles reconstruction is used to extract the initial skeleton of the vessel and obtain the local information of the vessel mask. The scale-space particle sampling method exploits the theory of linear scale-space to localize the features of the image described by the Hessian. The vessel tree is represented as a set of particles, each of which contains vessel scale, orientation, and intensity information. Therefore, particles are represented as $X = \{x_i\}$, where $x_i$ represents a particle point. However, after vessel tree reconstruction, the vessels are discontinuous due to the inability of this method to identify non-tubular structures, such as the bifurcation of vessels, resulting in local point loss. Additionally, there is no parent-child relationship between the particles.

Secondly, the skeleton extraction algorithm based on distance transform can maintain the connectivity of vessel tree. The MSFM is used to extract information about global connectivity from the vessel mask. The potential terminal points and the root node are obtained by calculating the distance map. Then, the vessel trajectory is obtained by iteratively tracing from the terminal points to the root node. During this process, the branch online confidence score is calculated in the time map to determine whether the trace iteration should be updated or stopped. In this article, we take the 3D vessel mask as input, aim to output the vessel skeleton tree G, and assign a 3D spatial coordinate and radius to each vessel skeleton tree G node. The degree of each vessel tree node can be between 1 and 3. However, the radius of each node of the obtained vessel tree is only an approximation and does not really reflect the vessel tubular shape. There are some differences between the reconstructed results and the real vessel tree.

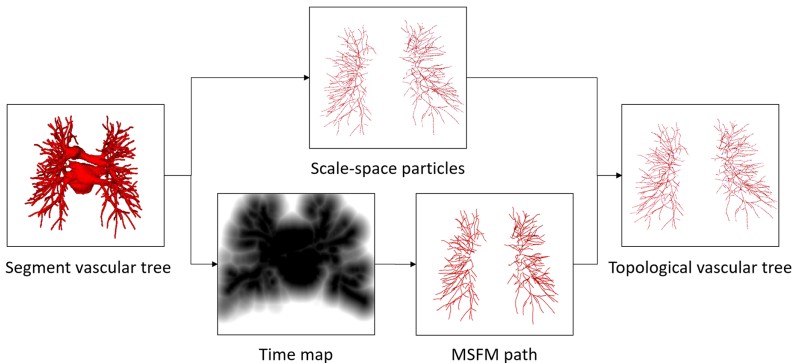

**Figure 3 The proposed vessel tree topology extraction process.** The initial topology is obtained by the scale-space particles, and the final vessel tree topology is obtained by refining the initial topology under the guidance of MSFM.               

Finally, in order to ensure the continuity and authenticity of the topology reconstruction, we flexibly combined the global and local information of the vessel skeleton. In this article, a particle-based 26-neighborhood search method is adopted to extract the topological structure of the vessels by guiding the time map to make up for the lost vessel particles. vessel particles can be classified into three categories: terminal points (or false-positive terminal points), branching points, and bifurcating points. We indicate the number of points in the 26 neighborhoods of vessel particles $x_i$ as $\Omega_{26}(x_i)$. The values of $\Omega_{26}(x_i)$ is different based on the type of particle $x_i$. For the terminal points, $\Omega_{26}(x_i) = 1$. Branching points are assigned $\Omega_{26}(x_i) = 2$. Bifurcating points have $\Omega_{26}(x_i) > 2$. Further discrimination is needed due to the presence of branch point loss or branch fracture resulting in false positive terminal points. Travel in the direction of the terminal vessel points, and if the terminal point is still on the vessel mask after traveling, it is considered a false positive terminal point, where the travel distance is between one and two scales. Otherwise, it is considered the true terminal point. For the false positive terminal points, the trajectory of the lost vessel points is obtained by MSFM in the time map, in which the time map is calculated from the 3D distance map. Finally, the complete topology tree is obtained. Based on the extracted topology tree, the information from each particle is used to construct the graph structure we need. Through the category of particle points and the parent-child node relationship between particles, we redefined a graph T = $\{X, \varepsilon\}$ composed of nodes X = $\{x_i\}$ and edges $\varepsilon = \{\varepsilon_{ij}\}$. Node are defined as $\varepsilon_{ij} = 1$, when $\Omega(x_i) = 2$. The process of the vessel tree topology extraction method is shown in Fig. 3. And compared with the scale-space particles, the advantages of the vessel tree topology extraction in this article are shown in Fig. 4.

## Twin-pipe network

The twin-pipe network is designed to improve the preliminary classification accuracy by learning the differences in A/V characteristics caused by different scales. One pipeline network is trained on the full vessel graph to learn image information and global connectivity, while the other pipeline network is specifically trained on the tubule vessel graph. Taking into account the physiological feature of small arteries accompanying the

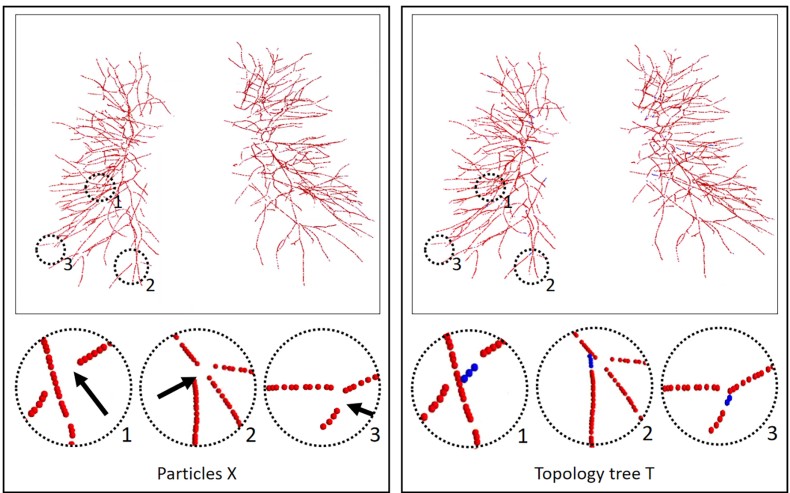

**Figure 4 Results of the vessel tree topology extraction method.** The left column is the result of the scale-space particles, and the right column is the result of our proposed vessel tree topology extraction, and the circles highlight the local skeleton points extracted from the topology. Black arrow points to the regions where the topology points are lost, and blue topology points indicate that our method makes up for the lost points. The local circular display position has been translated, rotated, and enlarged from the original image position.                               

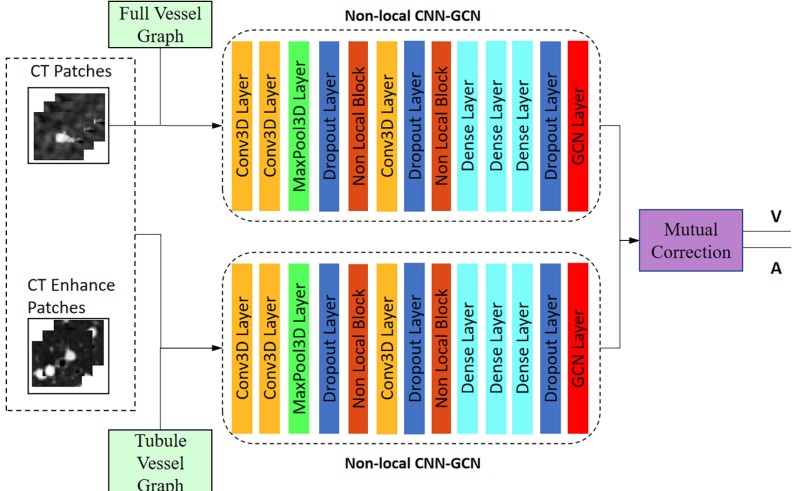

**Figure 5 Architecture of the twin-pipe network for preliminary A/V separation.**

small trachea, the twin-pipe network incorporates the CT original images and the vessel enhancement images as input patches. This enables the network to learn additional distinguishing features of pulmonary arteries and veins. Finally, the preliminary classification results are obtained through the mutual correction module.

The network structure is shown in Fig. 5. Twin-pipe network is made up of two non-local CNN-GCN classifiers. Traditional CNN networks have limitations in capturing long-range dependencies that extract the global understanding of visual scenes. In this article, we consider both image information and connectivity information by connecting Non-

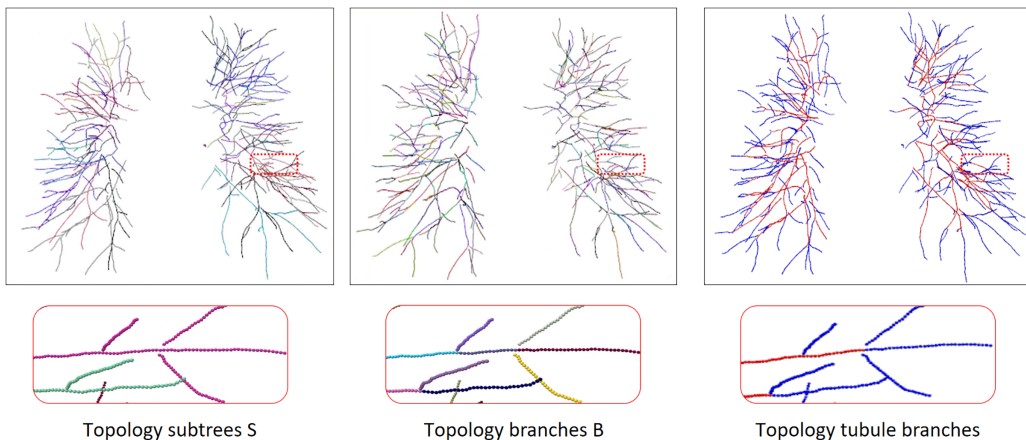

|  |  |  |
| :---: | :---: | :---: |
| Topology subtrees S | Topology branches B | Topology tubule branches |

**Figure 6 The process of topology tubule branch extraction, the columns from left to right are topology subtree, topology branch, topology tubule branch.** The different color segments represent different categories. Due to the complex and changeable structure of the vessel tree, the display position has been translated, rotated, and enlarged from the original image position.

local CNN and graph convolutional network (GCN). The Non-local CNN (*Sun et al., 2020*) considers the global context information of the image, while the GCN module (*Selvan et al., 2020*) can learn the local and graph connection information. Combining these two modules together is useful for analyzing the vessel tree. In order to connect the Non-local CNN with GCN, this article adapts the end-to-end method of CNN connecting GCN proposed by *Zhai et al. (2019)*. The definition of the GCN layer is as follows:

$$H^{(l+1)} = \sigma(WH^{(l)}\Theta^{(l)}) \tag{1}$$

The input of layer (L + 1) is the output of layer (L), the input size of $H^{(l+1)}$ is $N \times F^{(l+1)}$, and the size of output $H^{(l)}$ is $N \times F^{(l)}$, where W is the weight matrix of nodes X. $\Theta^{(l)}$ is a hierarchical training parameter matrix, $\sigma(\cdot)$ is an activation function, such as ReLU function. Image feature matrix Y, the output of Non-local CNN network, is taken as the input of GCN layer.

$$H^{(l+1)} = \sigma(WY\Theta^{(l)}), where\ Y = \Phi(P|\theta) \tag{2}$$

P is the local image patches centered around each particle, oriented perpendicularly to the blood vessel. Each image patch with the size of S = [32, 32, 3] is labeled as artery or vein according to the center voxel, *i.e.*, $y_i \in \{1, 2\}$. $\Phi()$ is a general Non-local CNN layer. $\theta$ is the Non-local CNN network training parameter.

To train the twin-pipe network, a set of n nodes and their corresponding image patches are randomly selected from graph T. These selected nodes and patches are represented as N with a size of n * S. The image patches of their neighbors are also required as we use GCN networks. Neighbor patches are denoted by MN which has the size m × n × S, where m is the number of neighbors. N and the neighborhood MN are the input of the network. We can extract each branch subgraph from graph T, and each branch subgraph is either an

artery or a vein. The tubule vessel graph is the set of tubule branches in the full vessel graph, we perform vessel enhancement and normalization on the original image to enhance the differences between arterioles and small trachea. The extraction process is shown in Fig. 6. The twin-pipe network is trained and predicts whether each central voxel is an artery or a vein.

## Topology optimizer

In order to separate arteries and veins, topological subtrees and topological branches are extracted from the topological tree T. Topological subtrees S are roughly extracted from root nodes through the uniform distribution of arteries and veins, and then refined according to tubular features and scale information. Specifically, extract the subtree root node in T, and then traverse each subtree root node until it reaches the terminal points to obtain the corresponding subtree. However, due to the presence of arterial and venous interlacing, it is possible that arterial subtrees contain venous branches, which is inevitable. The topological branch B is extracted from the backtracking path of the endpoints. In this process, the bifurcation points and terminal points are identified as the endpoints, ensuring that each branch is either an artery or a vein. The results of a topology subtree and topology branch are shown in Fig. 6.

Though the twin-pipe network can learn local, global, and connectivity information, spatial inconsistency may still occur during classification. Therefore, after the preliminary classification of the twin-pipe network, we apply a topology optimizer based on the vessel tree structure to refine the classification. First of all, for the local patch corresponding to each particle, the preliminary probability is given by the twin-pipe network. If the probability is greater than 0.5, the node is an artery. Otherwise, it is a vein. Then, the subtree $s_i$ is obtained from the extracted topological subtree S. The categorization of each subtree (artery or vein) is determined by the number of particles within the subtree that are predicted to be arteries or veins. Finally, the branch confidence of each subtree is calculated. When the subtree category and the branch category are inconsistent, and the branch has higher confidence, appropriate pruning is performed to correct the predicted results. The confidence calculation depends on the initial number of predicted arteries and veins in the subtree or branch.

In order to reconstruct the pulmonary artery and vein tree, based on the prediction result of the central particle and the scale information of the corresponding topological point, the vessel voxels in the scale space area of the topological point are labeled to belong to a category. It is then fused with vessels near the hilum for final A/V tree results.

## EXPERIMENTS

### Data description

The CT data in this article consist of three categories: Siemens-Non-16, GE-Non-8, and GE-A-8, these datasets are from a well-known local first-line 3A hospital. Our method was trained and validated on the Siemens-Non-16 dataset, and ablation experiments are also performed. Then, using GE-Non-8, GE-A-8 dataset, and the publicly available dataset CARVE14 for generalization experiments. The three types of data were selected because

the imaging quality of different devices and the image intensity under a contrast agent at different periods had an uncertain influence on pulmonary A/V separation. We describe three types of datasets in detail as follows:

1) Siemens-Non-16 datasets, a total of 16 cases, in which CT data were obtained by a Siemens scanner without a contrast agent.
2) GE-Non-8 datasets, a total of eight cases, in which CT data were obtained by a GE scanner without a contrast agent.
3) GE-A-8 datasets, a total of eight cases, in which CT data were obtained by a GE scanner with a contrast agent flowing through the artery.

The axial images of these CT data were resampled to $512 \times 512$ sizes, and the slice thickness varied from 0.625 to 1 mm. To more accurately evaluate the effectiveness of our method, manual A/V annotations repeatedly confirmed by two experienced chest experts were used as the reference standard for evaluation.

For the purpose of data equalization, eight out of the 10 fully annotated noncontrast CT scans from the public CARVE14 datasets were randomly selected to verify the generalization of our method. The CT data was from a Philips scanner with an axial image reconstruction of $512 \times 512$ sizes and a thickness of 1.0 mm.

## Evaluation metrics

To evaluate the effectiveness of the proposed method, we compare our method with the manual reference standard. The A/V separation effect is evaluated by the predictive accuracy of the vessel skeleton particles as an indicator. For this purpose, we calculate the accuracy of vessel particles, the percentage of correct A/V predictions.

$$Accuracy = \frac{TP + TN}{TP + FN + TN + FP} \tag{3}$$

Then, the sensitivity and specificity (considering arteries as the positives) are also computed, indicating the prediction accuracy of arterial or venous, respectively. Also the Dice Similarity Coefficient (DSC) is used to evaluate the effectiveness of the method.

$$Sensitivity = \frac{TP}{TP + FN} \tag{4}$$

$$Specificity = \frac{TN}{TN + FP} \tag{5}$$

$$DSC(P, G) = \frac{2 \times |P \cap G|}{|P| + |G|} \tag{6}$$

where P and G denote the predicted mask and ground truth, respectively. arterial points are positive samples, and venous ones are negative samples, TP, TN are the number of correctly predicted arterial and venous points; FP, FN are the number of incorrectly predicted arterial and venous points. In all experiments, the accuracy rate is considered the main evaluation measure. The sensitivity, specificity and DSC are also analyzed to complete the evaluation of the experiments.

## Experimental setup details

Training was performed by running on the Windows10 with Intel Core i7-9700 CPU (3.00 GHz), 32 GB RAM, and NVIDIA RTX 2080 GPU with 8 GB of memory. The twin-pipe network used the deep learning framework "Tensorflow.Keras" with version 1.12.0. In the training process, we used the SGD optimizer with a momentum of 0.9 and cross-entropy loss function for 150 epochs. The learning rate was 1e−3, and the batch size was 128.

## RESULTS

The evaluation structure of the method in the article is as follows. First, this article presents the results of our method on the Siemens-Non-16 dataset, as well as the results of the A/V separation methods in recent years. Second, the ablation experiments are performed on the Siemens-Non-16 dataset to validate each component of the proposed method. In addition, the generalization experiments are carried out to verify the proposed method performance from the GE-Non-8 dataset, the GE-A-8 dataset, and the CARVE14 dataset.

### Comparison with recent A/V separation methods

Table 1 presents the pulmonary A/V separation study results in recent years. As shown in Table 1, *Yu et al. (2021)* obtained an average accuracy of 85% in pulmonary A/V separation based on subtree relationships and anatomical knowledge. Under the voxel-based evaluation system, *Payer et al. (2016)* implemented A/V separation on enhanced CT based on two integer programming with an interactive accuracy of 96.3%. *Charbonnier et al. (2015)* used tree partitioning and peripheral vessel matching to classify arteries and veins with a median accuracy of 89.0% on noncontrast CT. *Qin et al. (2021)* designed end-to-end A/V separation method, which was directly realized by learning tubule-sensitive CNNs. In addition, no presegmentation or postprocessing was designed in the pipeline to avoid error accumulation, and this method reached 90.3% in noncontrast CT. Under the particle-based evaluation system, *Zhai et al. (2019)* proposed a new network for end-to-end training, which greatly reduced the algorithm complexity and its dependence on parameters. The proposed CNN-GCN method improved lung A/V separation was compared with the baseline CNN method. *Nardelli et al. (2018)* used deep learning to solve parameter optimization and automatically learn the difference between pulmonary A/V. The overall accuracy of this method reached 94% in noncontrast CT. *Jimenez-Carretero et al. (2019)* demonstrated a significant improvement in separation on noncontrast CT using the graph-cut method. Compared with the manual reference standard, the proposed A/V separation method in this article achieved an average accuracy of 96.2% and a DSC of 81.7% on the Siemens-Non-16 dataset. It can be seen that the accuracy of the proposed method for A/V classification is significantly improved compared to other methods. But at the same time we also note that the relatively small improvement in the DSC can be attributed to the limitations of using DSC as a quality metric for tubular and curved structures. The Dice coefficient tends to focus on accurately assessing the main large vessels rather than ensuring the global topological connectivity of the branch vessels (*Banerjee et al., 2022*).

**Table 1 An overview of the results of recent methods.**

|  | DSC | Accuracy | Evaluation method | Data type and cases |
|---|---|---|---|---|
| *Yu et al. (2021)* | — | 85.0% | Branch number | CT-10 |
| *Payer et al. (2016)* | — | 96.3% | Voxel | Enhanced CT-25 |
| *Charbonnier et al. (2015)* | — | 89.0% | Voxel | Noncontrast CT-55 |
| *Qin et al. (2021)* | 82.4% | 90.3% | Voxel | Noncontrast CT-55 |
| *Zhai et al. (2019)* | 75.2% | 77.8% | Particle | Enhanced CT-11 |
| *Nardelli et al. (2018)* | — | 94.0% | Particle | Noncontrast CT-21 |
| *Jimenez-Carretero et al. (2019)* | — | 89.1% | Particle | Noncontrast CT-48 |
| Our approach | 81.7% | **96.2**% | Particle | Noncontrast CT-16 |

Note:
For each metric, the top-performing method is shown in bold.

**Table 2 Preliminary classification results of a baseline network (Non-local CNN-GCN network) and twin-pipe network presented in this article.**

|  | Acc. (%) | Sens. (%) | Spec. (%) |
|---|---|---|---|
| Baseline in full vessels | 91.6 | 88.4 | 94.1 |
| Baseline in tubule vessels | 91.2 | 86.5 | 94.8 |
| Twin-pipe in tubule vessels | **91.8** | 88.1 | **94.7** |

Note:
For each metric, the top-performing method is shown in bold.

## Ablation experiments

We investigate the effectiveness of the key components of the proposed method, including the twin-pipe network and topological postprocessing optimization.

### Twin-pipe network

To verify the effectiveness of the proposed twin-pipe network, we trained and tested on the Siemens-Non-16 dataset and compared and analyzed the preliminary classification accuracy of tubule vessels using the proposed twin-pipe network and baseline Non-local CNN-GCN network. As shown in Table 2, the classification accuracy, sensitivity, and specificity of the baseline network on the tubule vessels were 91.2%, 86.5%, and 94.8%, respectively. By comparison, the classification accuracy, sensitivity, and specificity of the proposed twin-pipe network on the tubule vessels were 91.8%, 88.1%, and 94.7%, respectively. That is, the twin-pipe network outperformed the baseline network. The experimental results indicate that the baseline network has lower accuracy in classifying tubule vessels at the distal end compared to classifying full vessels. This is primarily because the features of tubule vessel branches are noticeably inconsistent with those of the main vessel branches, and they are susceptible to the influence of tubule bronchial branches and partial volume effects. However, the use of a twin-pipe network approach effectively enhances the accuracy of classifying distal tubule vessels. In other words, the twin-pipe network outperforms the baseline network in terms of performance. It effectively overcomes the impact of different scales, particularly in learning the closely related information of distal bronchial and arterial branches.

**Table 3 Overview of the results obtained compared to other postprocessing refinement strategies.**

|  | Acc. (%) | Sens. (%) | Spec. (%) |
|---|---|---|---|
| Based on particle | 91.8 | 89.1 | 94.1 |
| Based on branch | 95.3 | 93.2 | 96.9 |
| Based on subtree | 93.2 | 92.2 | 94.1 |
| Our approach | **96.2** | **94.1** | **97.8** |

Note:
   For each metric, the top-performing method is shown in bold.

**Table 4 Overview of the results obtained under different types of CT scans.**

| DataSet | Method | DSC. (%) | Acc. (%) | Sens. (%) | Spec. (%) |
|---|---|---|---|---|---|
| The GE-Non-8 | Our approach | 80.7 | 93.8 | 92.0 | 95.5 |
| The GE-A-8 | Our approach | 81.4 | 94.8 | 91.8 | 97.7 |
| CARVE14 | *Charbonnier et al. (2015)* | — | 89.0 | — | — |
|  | *Qin et al. (2021)* | — | 90.3 | — | — |
|  | Our approach | 78.8 | 90.1 | 90.7 | 89.8 |

### *Topology optimizer*

To prove that the proposed topology optimizer is reasonable, we design different topology strategy optimizers to compare and analyze the experimental results. This approach includes the following: the twin-pipe network precision (based on particle), the topology branch refining precision (based on branch), the topology subtree refining precision (based on subtree), and the proposed topology optimizer refinement precision. Table 3 shows an overview of the accuracy under different topology strategy optimizers and reports sensitivity and specificity. The lowest accuracy is based on particles due to the possible spatial inconsistency of the separation results. Therefore, topological sub-trees or topological branches constraint need to be used to maintain consistency. However, due to A/V interleaving, there may be inconsistencies between branch categories and the subtree categories they belong to, which may limit its effectiveness in achieving higher accuracy. The topology optimizer we propose outperforms other approaches in accuracy by calculating the calculated branch confidence and correcting incorrect branches.

## Generalization experiments

To verify the generalization of the proposed method, we performed validation on three datasets of different types: the GE-Non-8 dataset from another device GE, the GE-A-8 dataset from another arterial enhancement model, and the publicly available CARVE14 dataset. Table 4 shows the summary of case results under different types of CT scans, The proposed method achieved an accuracy of 93.8% and a DSC of 80.7% on the GE-Non-8 dataset. Similarly, under the influence of dynamic changes in the arterial enhancement effect with time and contrast agent dose, the characteristics varied, and the difficulty of A/V separation increased. In the GE-A-8 dataset, the proposed method still achieved an accuracy of 94.8% and a DSC of 81.4%. These results proved that the proposed method

maintained high classification accuracy under different devices and modes. In addition, we verified the proposed method on the public CARVE14 dataset, and two proposed artery-vein classification methods were evaluated: *Charbonnier et al. (2015)* and *Nardelli et al. (2018)*. Due to the difficulties in annotating this dataset, some voxels were annotated as uncertain vessels. Moreover, pulmonary vessels near the hilum were not annotated, which resulted in a lack of continuity in the vessel tree. As a result, there exists a gap between these annotations and clinical applications. Thus, some critical components of the proposed method were used for simple prediction, and a competitive classification accuracy of 90.1% and a DSC of 78.8% was obtained. With the experimental results, we clearly demonstrate the accuracy and generalization ability of the proposed method for different datasets and situations. These results support the effectiveness and feasibility of the proposed method.

## DISCUSSION

The proposed end-to-end framework constructs topological trees by extracting vessels from CT and obtains preliminary classification results through a twin-pipe network. Subsequently, a topological optimizer is used to refine the A/V classification results and reconstructs pulmonary A/V through the scale information at last. An objective inter-work comparison could be difficult due to the difference in medical data sets and implementation, so our experiments are mainly conducted on our own CT data. The experimental results in Table 1 indicate that the average accuracy of the proposed method can reach 96.2% compared with that of manual reference annotation. The experimental results in Table 4 show that our method is also applicable to CT images from different devices and different modes, thus the proposed method has good generalization.

Pulmonary A/V separation is a challenging problem in medical image analysis. The separation ought to overcome the complexity of the pulmonary structure as well as the relatively limited resolution of CT images. Our evaluation method for A/V separation is mainly based on topological particles rather than voxel classification. The reason is that in clinical practice, surgeons focus more on the structural branching direction of vessels, consistent with the evaluation system based on topological particles. A/V separation provides effective information for surgical planning and navigation, while proximal vascular extension aids surgeons in locating vessels more quickly. Therefore, the final A/V separation results include vessels close to the hilum of the lung, as shown in Fig. 7. Our evaluation system for the A/V separation method does not include vessels near the hilum, mainly because the vessels near the hilum are abnormally large and non-tubular, and the vessel topology could not be extracted.

First, we extract vessel topology for the training of the classification network. Currently, some topology methods such as thinning (*Palágyi & Németh, 2017*), geodesics and shortest paths (*Chen, Mirebeau & Cohen, 2016*), and some other special methods (*Liu et al., 2017*) are used to construct vessel topology. However, validating these methods with accuracy is difficult, mainly because (1) it is difficult for these methods to obtain manual annotations based on high-resolution images, and (2) these methods pay more attention to vessel branch direction, bifurcation pixel redundancy, loopback, fracture, *etc.*, We propose a

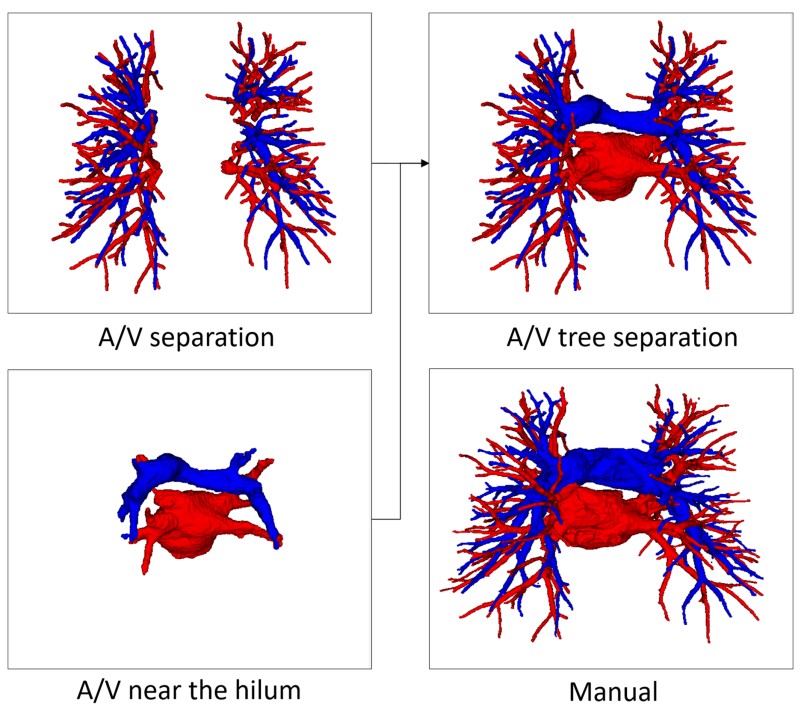

A/V separation            A/V tree separation

A/V near the hilum            Manual

**Figure 7** **An example of manual separation and the A/V separation presented in this article.**

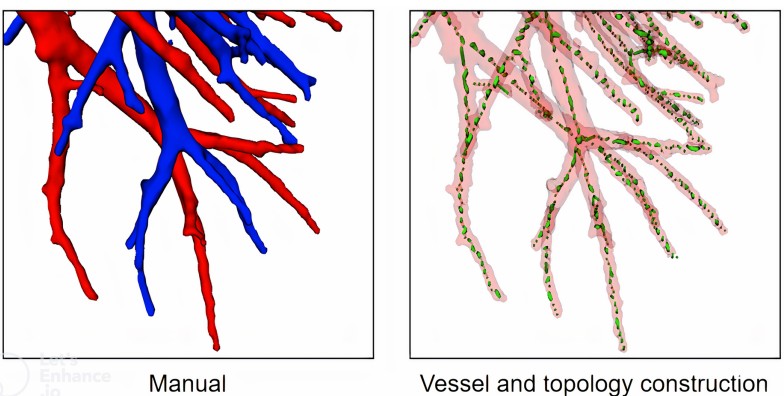

Manual            Vessel and topology construction

**Figure 8** **A case of local vessel and topology construction.**

vessel tree topology method that fully uses the advantages to solve the topological fracture problem caused by particle loss. As shown in Fig. 8, topological skeleton nodes are basically located on the central axis of vessels, and there is no obvious pixel redundancy or fracture at the bifurcation of vessels.

Second, we design a twin-pipe network for the preliminary classification of A/V and to improve the classification accuracy of tubule vessels. As shown in Table 2, the classification accuracy of tubule vessels in the baseline network is lower than that of the full vessels, while the sensitivity of the twin-pipe network in the tubule vessels is higher than that of the

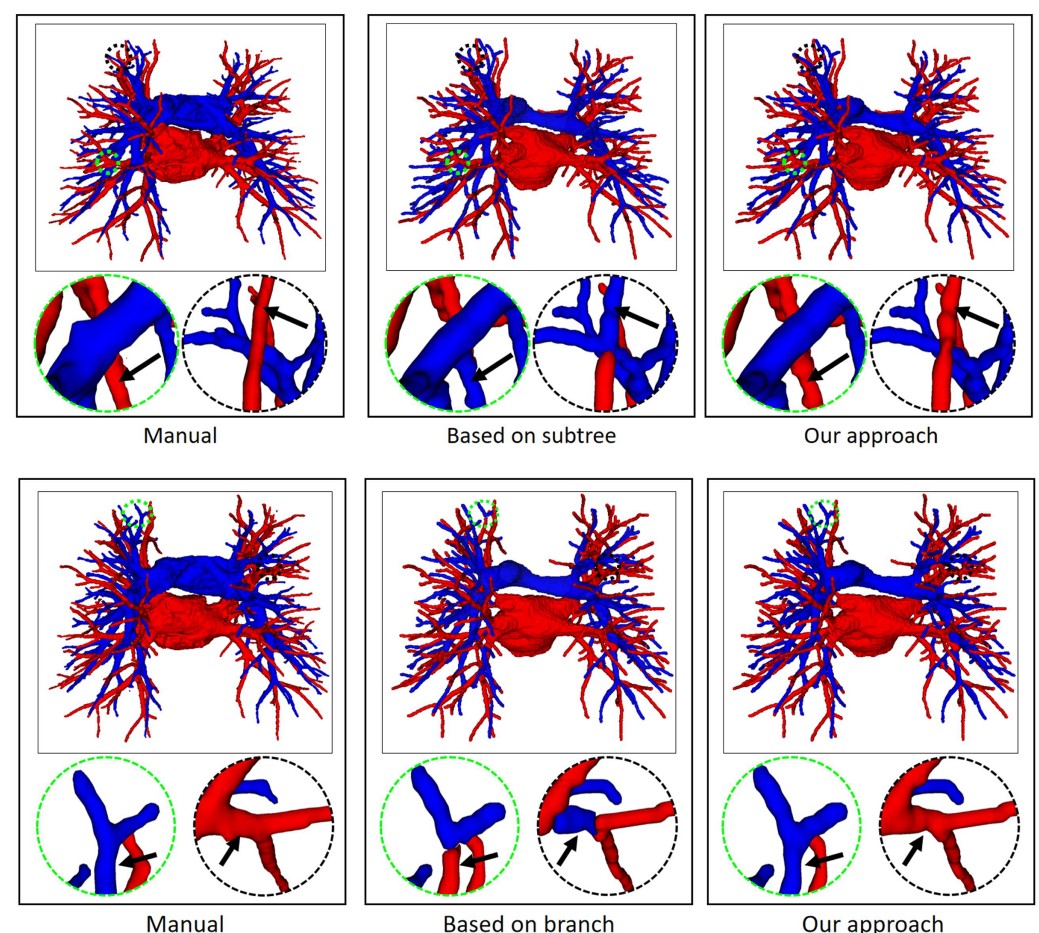

**Figure 9 In an example of A/V separation results, the reconstructed results of different topology strategy optimizers.** The circular area highlights local classification results. Due to the complex and changeable structure of the vessel tree, the local circular display position has been translated, rotated, and enlarged from the original image position.               

baseline network. The experimental results show that the error frequency is lower in the tubule vessels in the twin-pipe network. That is, the twin-pipe network can effectively improve the tubule vessel classification accuracy. The experiment proves that: (1) the Non-local CNN-GCN network can effectively combine local information, global image information, and graph connection information. Differences in A/V characteristics can be automatically learned, eliminating the need for complex parameter tuning. (2) The twin-pipe network effectively captures the differences in A/V characteristics at various scales and learns the close association between the tubule bronchus and the artery. It is superior to the baseline Non-local CNN-GCN network and does not depend on the airway segmentation results.

Finally, the topology optimizer extracts the topology of the subtrees and their branches' refinement results using the method in "Topology Optimizer". Then, we use topology subtrees and topology branches for postprocessing. As shown in Table 3, the proposed topology optimizer accuracy is superior to that of the subtree-based and branch-based

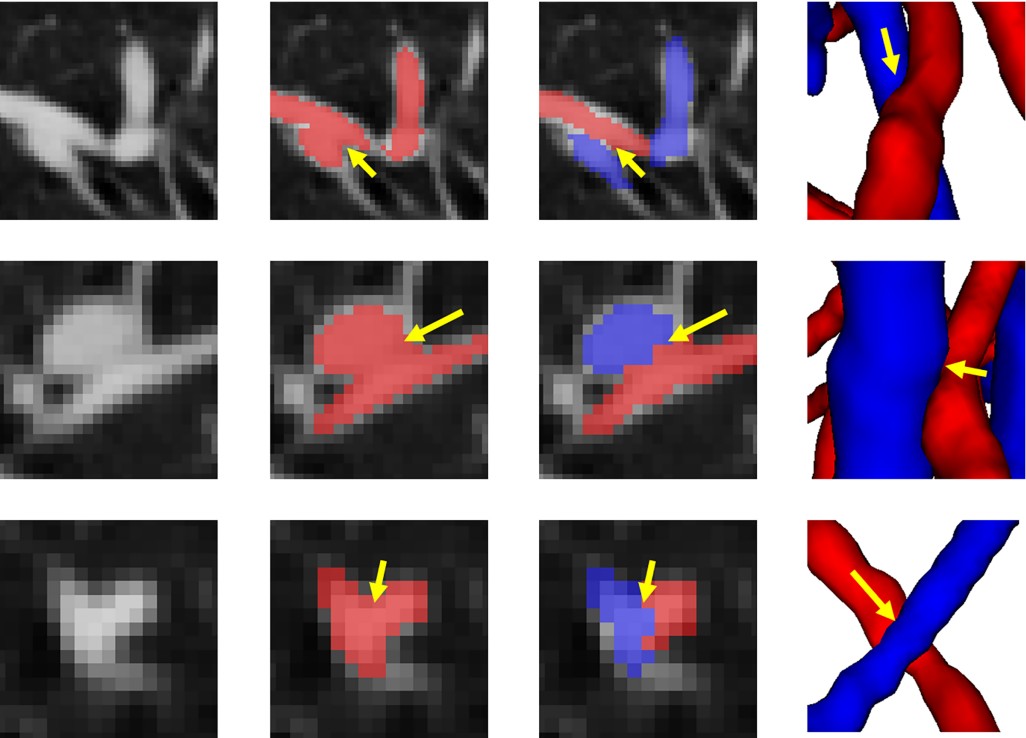

**Figure 10 The results of separation of arterial and venous intersections are shown in this figure.** Each row represents a different location where arteries and veins intersect. The number of columns respectively represents the location of the intersecting arteries and veins in the original CT image, the results of vessel segmentation, the results of the A/V separation method presented in this article, and the results of the 3D reconstruction of this location. The blue areas represent the arteries and the red veins.

topology optimization. Fig. 9 shows the reconstructed results of different topology strategy optimizers. The results in the first row show that when the number of points on the branch is small, the branch-based topology optimization method is prone to prediction errors. This is owing to the branch-based refinement strategy focusing on the relationships within the branches and ignoring the topological relationships between branches. The second row shows that topology optimization based on the subtree strategy is prone to prediction errors in the case of A/V intersection. Due to the complex intertwining of arteries and veins trees, the points where arteries and veins intersect are often mistaken for arterial subtree bifurcation points. As a result, venous branches are misclassified as arterial subtrees, leading to classification errors.

Figure 10 shows the final separation result of our method at the intersections of the arteries and veins. The proposed topology optimizer considers the interbranch and intrabranch relationships. The extracted topological subtree is used to maintain the spatial connectivity of topological particles. The branch confidence calculated using topological branches is used to correct the topological subtree. To some extent, our method can solve the misclassification problem caused by interlaced arteries and veins. This method made some achievements in A/V separation, but there are still some unavoidable shortcomings. The method is based on topological particle reconstruction of A/V classification, and some

errors are found between the reconstructed vessel size and the actual situation. Although it is possible to classify the A/V at the intersections, the scale of the intersections after reconstruction is abnormal and often results in fractures or abnormal expansion. This condition occurs because topological particles at non-tubular intersections of arteries and veins cannot be accurately extracted during topology construction, thereby leading to topological particle scale mutation or deletion. To apply it to clinical practice, we can further improve the classification results and performance by optimizing the topology extraction algorithm and detecting the intersection situation, as well as increase the clinical applicability through semi-automated graphical user interface interaction operations.

## CONCLUSION

In this article, we propose a novel method for automatically separating pulmonary arteries and veins. This method is applicable for noncontrast CT scans that lack vessel edge information, and by incorporating vessel topology information, it can resolve adhesions of arteries and veins and misclassification of small vessels. We extract the vessel skeleton through the vessel tree topology method, obtain the preliminary classification results after twin-pipe network training, and use the topology structure information for postprocessing. Our approach has been tested on the public dataset CARVE14 and our private dataset. Experimental results show that the proposed method has good performance and generalizability to CT images from different devices and different modes. The method proposed in this article will better assist physicians in the planning of surgery for lung disease.

### Funding
This work was supported by the Fujian Provincial Natural Science Foundation project (Grant Nos. 2021J02019, 2020J01472, 2020Y9091, 2022J01082, 2022Y4014, 2022L3003). The funders had no role in study design, data collection and analysis, decision to publish, or preparation of the manuscript.

### Grant Disclosures
The following grant information was disclosed by the authors:
Fujian Provincial Natural Science Foundation Project: 2021J02019, 2020J01472, 2020Y9091, 2022J01082, 2022Y4014, 2022L3003.

### Competing Interests
The authors declare that they have no competing interests.

### Author Contributions
- Lin Pan conceived and designed the experiments, performed the experiments, authored or reviewed drafts of the article, and approved the final draft.
- Xiaochao Yan performed the experiments, performed the computation work, prepared figures and/or tables, and approved the final draft.

- Yaoyong Zheng conceived and designed the experiments, performed the experiments, performed the computation work, prepared figures and/or tables, and approved the final draft.
- Liqin Huang conceived and designed the experiments, authored or reviewed drafts of the article, and approved the final draft.
- Zhen Zhang analyzed the data, performed the computation work, prepared figures and/or tables, and approved the final draft.
- Rongda Fu analyzed the data, performed the computation work, prepared figures and/or tables, and approved the final draft.
- Bin Zheng analyzed the data, authored or reviewed drafts of the article, and approved the final draft.
- Shaohua Zheng conceived and designed the experiments, authored or reviewed drafts of the article, and approved the final draft.

## Data Availability

The code is available in the Supplemental File and the dataset is available at Zenodo and https://arteryvein.grand-challenge.org/:

van Ginneken, Bram. (2019). ANODE09 [Data set]. Zenodo. https://doi.org/10.5281/zenodo.3595212.

## Supplemental Information

Supplemental information for this article can be found online at http://dx.doi.org/10.7717/peerj-cs.1537#supplemental-information.

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
