# Peer review of "Automatic pulmonary artery-vein separation in CT images using a twin-pipe network and topology reconstruction"

_PeerJ Computer Science, doi:10.7717/peerj-cs.1537_

## Round 0.1 · original submission · Major Revisions

You have some reviews with valuable comments that will help improve the work submitted to PeerJ Computer Science. Review what the reviewers have indicated, try to adjust what you consider appropriate, or rebut what you do not believe is suitable.

·

Basic reporting

No comment

Experimental design

No comment

Validity of the findings

No comment

Additional comments

The paper aims to propose the twin-pipe network and topology reconstruction method for automatically separating pulmonary arteries and veins trained and validated on the in-house dataset and tested on CARVE14. The paper can be of interest to the community. However, to make it publishable, I have some comments and suggestions as follows.
1) The English language should be improved to ensure that an international audience can clearly understand your paper. Some sentences are so long and complicated – the current phrasing makes comprehension difficult.
2) More clarifications and highlights about the research gaps are in the related works section.
3) The quality of Figure 2 should be improved. For example, when zooming in on the picture, the font will blur.
4) What is the patch size of the CT image in the stage of vessel tree extraction using 3D UNet?
5) Results need more explanations. Additional analysis is required at each experiment to show its main purpose.
6) Add more evaluation metrics, for example, Dice.

Reviewer 2 ·

Basic reporting

This paper presents an automated framework for classifying arteries and veins from CT images. The manuscript is effectively written and maintains a clear flow of information. The experiments conducted are robust, demonstrating the reliability of the proposed approach. The level of novelty and originality exhibited in this work is commendable. Overall, the paper only requires minor revisions.

Experimental design

The research question is clearly defined and holds significant relevance, offering valuable contributions to the scientific community's existing body of work. The proposed technique appears promising in terms of its potential to enhance the current state of the field. Furthermore, the inclusion of multiple experiments to validate the proposed method against other state-of-the-art techniques is commendable. However, it is suggested that in table 4, which presents results on the CARVE14 dataset, a comparison to other techniques such as Charbonnier et al. and Nardelli et al. should be included. This additional information would provide a better understanding of the performance of the proposed method in relation to its counterparts.

Validity of the findings

As stated above, a comparison to other techniques should be provided in table 4.

---

## Round 0.2 · accepted · Accept

We are delighted to inform you that your manuscript entitled has been accepted for publication in PeerJ Computer Science.

We would like to extend our heartfelt congratulations on successfully addressing and fulfilling all the valuable comments and suggestions provided by our esteemed reviewers. Your dedication and commitment to refining the manuscript have resulted in a significant improvement in the overall quality and clarity of the work.

·

Basic reporting

No comments

Experimental design

No comment

Validity of the findings

No comment

Additional comments

Dear Author,
Thank you for the revised manuscript entitled "Automatic pulmonary artery-vein separation in CT images using a twin-pipe network and topology reconstruction" I appreciate the efforts you have made in addressing the comments.